# Chemical, Pharmacological and Computerized Molecular Analysis of Stem’s Extracts of *Bauhinia scandens* L. Provide Insights into the Management of Diarrheal and Microbial Infections

**DOI:** 10.3390/nu14020265

**Published:** 2022-01-09

**Authors:** Md Minarul Islam, Rashedul Alam, Hea-Jong Chung, Nazim Uddin Emon, Mohammad Fazlul Kabir, Sajib Rudra, Safaet Alam, Ahsan Ullah, Seong-Tshool Hong, Mohammed Aktar Sayeed

**Affiliations:** 1Department of Biomedical Sciences and Institute for Medical Science, Jeonbuk National University Medical School, Jeonju 54907, Korea; mislambcmb@gmail.com (M.M.I.); seonghong@jbnu.ac.kr (S.-T.H.); 2Department of Medical Science, Jeonbuk National University Medical School, Jeonju 54907, Korea; rashedcmu@gmail.com (R.A.); pharmahsan18@gmail.com (A.U.); 3Gwanju Center, Korea Basic Science Institute, Gwanju 61715, Korea; hjchung84@kbsi.re.kr; 4Department of Pharmacy, Faculty of Science and Engineering, International Islamic University Chittagong, Chattogram 4318, Bangladesh; nazim7emon@gmail.com; 5Department of Biology, Georgia State University, Atlanta, GA 30303, USA; Kabirrasel07@gmail.com; 6Department of Botany, Faculty of Biological Sciences, University of Chittagong, Chattogram 4331, Bangladesh; rudrasajibcu89@gmail.com; 7Department of Pharmacy, State University of Bangladesh, 77 Satmasjid Road, Dhanmondi, Dhaka 1205, Bangladesh; safaet.du@gmail.com

**Keywords:** *Bauhinia scandens*, UPLC-QTOF–M.S., entero-pooling, peristaltic index, phytochemicals, receptors

## Abstract

*Bauhinia scandens* L. (Family: Fabaceae) is commonly used to treat cholera, diarrhea, asthma, and diabetes disorder in integrative medicine. This study aimed to screen the presence of phytochemicals (preliminary and UPLC-QTOF–M.S. analysis) and to examine the pharmacological activities of *Bauhinia scandens* L. stems (MEBS) stem extracts. Besides, in silico study was also implemented to elucidate the binding affinity and drug capability of the selected phytochemicals. In vivo anti diarrheal activity was investigated in mice models. In vitro, antibacterial and antifungal properties of MEBS against several pathogenic strains were evaluated using the disc diffusion method. In addition, in silico study has been employed using Discovery studio 2020, UCFS Chimera, PyRx autodock vina, and online tools. In the anti-diarrheal investigation, MEBS showed a significant dose-dependent inhibition rate in all three methods. The antibacterial and antifungal screening showed a remarkable zone of inhibition, of the diameter 14–26 mm and 12–28 mm, by MEBS. The present study revealed that MEBS has remarkable anti-diarrheal potential and is highly effective in wide-spectrum bacterial and fungal strains. Moreover, the in silico study validated the results of biological screenings. To conclude, MEBS is presumed to be a good source in treating diarrhea, bacterial and fungal infections.

## 1. Introduction

Diarrhea is a chronic gastrointestinal disorder with potentially lethal consequences characterized by bowel movement, abdominal pain, and wet stool [1]. A significant cause of acute diarrhea is bacterial gastrointestinal (G.I.) infections. So far, it is challenging to identify particular pathogens in such situations. Dreadful complications can result from infection with various invasive microorganisms such as *Clostridium difficile*, *Salmonella* spp., *Campylobacter* spp., and *Shigella* spp. Diarrhea treatment is currently not considered precise and is typically intended to minimize the distress and irritation of frequent diarrhea. Conventional drugs used to treat diarrhea are not promising due to adverse effects: abdominal pain, constipation, dizziness, nausea or vomiting, blackened stools, tinnitus, etc. can be caused by these drugs [2]. Although several synthetic drugs such as Imodium, Kaopectate, and Pepto-Bismol have emerged over the years, none has formed a place in the ideal management of diarrhea [3]. Besides, many bacterial strains (gram-negative: *Campylobacter* spp., *Escherichia coli*, *Salmonella* spp., *Shigella* spp., *Vibrio cholera*, and gram-positive: *Bacillus cereus*, *Clostridium perfringens*, *Lactobacillus casei*, *Listeria monocytogenes*, *Staphylococcus aureus*, etc.) are responsible for numerous fatal diseases or dietary contamination [4,5]. Previous research has already demonstrated the antimicrobial action of plant extracts against bacteria [6]. Fungal spores are also widespread in the atmosphere, and fungal pathogens or their byproducts such as *Trichoderma* spp., *Candida* spp., *Aspergillus* spp., *Fusarium* spp. etc. also causes various illnesses. However, antifungal therapy plays an integral role in medication, and it is widespread to screen medicinal plants for the quest for new antifungal agents [7]. Substantially, the search for new antifungal agents largely relies on ethno-botanic and ethnopharmacological information [8]. Therefore, the response of traditional medication used as a modern type of drug, and microbial resistance to available antibiotics has driven specialists to explore herbal preparations for treating diarrhea, bacterial and fungal infections [9].

In such circumstances, plant-originated medicines might provide a comprehensive solution in anti-diarrheal medication. Hence, the WHO and other organizations have empowered investigations to treat and prevent diarrheal sicknesses relying upon conventional medicinal practices [10]. Plants are a formidable natural well-spring of bioactive phytochemicals from which various industrial chemicals have been obtained from time to time. In ancient educational history, Atharvaveda (Indian book), Ayurveda (Indian folk medication), and others have provided extensive information on preventive and therapeutic phytomedicines. The scope of phytomedicine containing antimicrobial, antifungal and anti-diarrheal content has generally been the domain of folk therapists [11]. Phytochemicals can be extracted from bark, flowers, fruits, leaves, roots, seeds and any other plant parts. In several laboratories worldwide, systematic sampling of plant species and their derivatives to explore bioactive compounds is routinely carried out. To discover novel biologically active compounds, research on herbal plants should be expanded. In addition, scientific experimentation of these therapies can lead to standardization of the quality and safety of these drugs in clinical use. These research events may also contribute to the production of sustainable and more persuasive drugs in the future [12]. In this manner, the quest for plants with anti-diarrheal and antimicrobial activities was of high interest to this study. 

*Bauhinia scandens* L. (Family: Fabaceae) is a chemically enriched medicinal plant that is generally consumed as a folk medicinal source. This plant contains 1-O-alkylglycerol, one of the most essential ether-lipids, existing in breast milk, spleen, bone marrow, red cells, blood plasma, uterine carcinoma, and liver [13,14]. Along with others, *Bauhinia scandens* is one of the most commonly used medicinal plants in Bangladesh. Local practitioners and ethnobotanical studies recommended that *Bauhinia scandens* has different medicinal properties against several clinical complexities such as Alzheimer’s disease, cardiovascular disease, tumor, acute dysentery, diarrhea, diabetes, fever, etc. [15,16,17]. This research framework reports the in vivo anti-diarrheal, in vitro antibacterial, and antifungal potency of the plant extracts. Subsequently, phytochemical screening and High-resolution UPLC-QTOF–M.S. have been accelerated to identify the phytochemical elements of methanol extracts of *Bauhinia scandens* L. stems (MEBS). Besides, in in silico studies, some identified phytochemicals were implemented to check their binding affinity to the M3 muscarinic acetylcholine receptor, human glutamate carboxypeptidase II (GCP II) E. coli exonuclease, I, GPCR beta-arrestin, and Cytochrome P450 14 alpha-sterol demethylase receptors.

## 2. Materials and Methods

### 2.1. Plant Material

Plant stems of the *Bauhinia scandens* were obtained in September 2019 from Kaptai (22°30′ N 92°13′ E), Rangamati, Bangladesh and the taxonomic identity was confirmed of the collected plant with the help of a botanist (accession number: CTGUH SR 7918). The specimen voucher has been deposited at the Department of Botany, University of Chittagong, Chittagong 4331, Bangladesh. Collected plant stems were washed thoroughly and dried under natural shade at 23 ± 2 °C for about 14 days. Further, plant materials were oven-dried at comparatively low temperature (40 °C) for an hour in order to be well ground. Following that, the dried stems were ground into a fine powder using a grinder (TCEP-FZ102). Powdered plant extracts were stored and preserved in a well-contained plastic box.

### 2.2. Extraction of the Plant Material

Around 500 g dry powder of the plant material was taken in a separated clean and dry glass compartment and soaked in methanol (2.5 L). The container with its materials was fixed with aluminium foil and a container lid and kept for 14 days at 23 ± 2 °C, with gentle shaking and mixing. Then the entire mixture was filtered with cotton, followed by no.1 Whatman double rings filter paper (Bibby RE200, Sterilin Ltd., Newport NP11 3EF, UK). The filtrates were incubated in a water bath at 40 °C in such a manner that the solvent could be evaporated, and 18 g crude methanol extract was obtained from the stems. The crude extract was stored at 4 °C in a refrigerator.

### 2.3. Drugs and Chemicals

To investigate anti-diarrheal efficacy, drugs and chemicals were purchased from reputed pharmaceutical companies of different countries: Castor Oil from WELL’s, Madrid, Spain, 10% charcoal in 5% gum acacia and Tween 80 from Sigma-Aldrich, St. Louis, MO, USA. Loperamide, amoxicillin and fluconazole were obtained from Beximco Pharmaceuticals Limited, Tongi, Bangladesh. The lab-grade chemicals to implement the phytochemical screenings were accrued from our laboratory. High-resolution UHPLC-QTOF–M.S. analysis was implemented at Malay Peninsula Standard, UKM, Malaysia.

### 2.4. Test Strains

In the antibacterial and antifungal study, all the chemicals and clinical pathogens were obtained from the microbiology laboratory, Department of Pharmacy, Faculty of Science and Engineering, International Islamic University of Chittagong, Bangladesh.

### 2.5. Animals

Swiss albino mice of around 25–30 g body weight and 7–8 weeks mature were collected from BCSIR (Bangladesh Council of Scientific and Industrial Research), Chattogram, Bangladesh. All the animals were kept in plastic cages at 20 ± 2 °C temperature and facilitated with a 12 h light-dark cycle and with the standard provision of food and plenty of water supplies. Isolated and silent conditions were maintained for all in vivo experiments. The protocols for directing the experimentations on the animal models were endorsed by the institutional ethical committee [18,19]. The Federation of European Laboratory Animal Science Associations (FELASA) recommendations and guidelines were employed to ascertain the reduction of pain and stress for the laboratory models. The animals were captivated and acclimatized to laboratory grade for 10 days before experimentation. Universally accepted rules and regulations were followed for the maintenance of experimental animals [20]. The protocols of the research were approved by the Planning & Development (P&D) committee of the Department of Pharmacy, International Islamic University Chittagong under approval no: 143/14-15/12/10/2019.

### 2.6. Phytochemical Screening

The phytochemical examinations of MEBS was implemented to demonstrate the presence of flavonoids, alkaloids, terpenoids, carbohydrate, saponins, tannins, glycosides, phloro-tannins, protein, phenolic, and steroids using previously established protocols [21].

### 2.7. High-Resolution UHPLC-QTOF–M.S. Analysis

The phytochemical screening of the methanol extract was performed using UHPLC-M.S. The UHPLC-M.S. study was conducted in conjunction with a Waters Xevo G2 Q-TOF mass spectrometer [22] employing Waters ACQUITY UHPLC IClass/Xevo (Milford, MA, USA). *B. scandens* sample extract was processed by dissolving 1 g of the plant extract in 1 mL of methanol. The separation was carried out on a 2.1 mm × 50 mm Zorbax Eclipse plus Acquity UHPLC BEH C18 (1.7 μm particle size). The UHPLC was configured with a Q-TOF mass spectrometer column with the sources of positive and negative electrospray ionization (ESI). Full-scan mode of UHPLC from *m*/*z* 50 to 1000 was executed with a source temperature of 120 °C. Solvent A was water, and solvent B was acetonitrile, both with 0.1% formic acid. Gradient elution was conducted for initial 15 min beginning with 99% solvent A and 1 percent solvent B and afterward 65% solvent A and 35 percent solvent B for 1 min, followed by a gradual increment of up to 100% for 2 min in solvent A and ultimately a gentle upsurge to 99% in solvent B and 1% in solvent A for 2 min. As a nebulizing and collision gas, highly pure nitrogen (N2) and ultra-highly purified helium (He) were implicated. The capillary voltage was fixed at 2.0 kV concerning the positive electrospray mode. Additional implied instrument parameters were: 100 V source offset; 550 °C desolvation temperature; 50 L/h cone gas flow at 120 °C temperature; and 800 L/h desolvation gas flow.

### 2.8. Acute Oral Toxicity

The MEBS (2000 mg/kg) was introduced orally to five male and five female mice for 14 days at the fixed-dose concentrations. The goal was to ascertain a dose that can generate morbidity as solid signs of toxic effects without mortality. Further lower and/or higher doses, and necessity for further experiments, were decided based on the first test results, e.g., mortality indicated obligatory retesting at a lower concentration dose. Food and water were given ad libitum to all animals for 72 h, and toxic symptoms and mortality were observed [23]. 

### 2.9. Anti-Diarrheal Assay

#### 2.9.1. Castor Oil-Induced Diarrhea in Mice

Previously described methods mentioned by Emon et al. [24] with slight modification were executed in this study. Initially, the mice were screened by feeding 0.5 mL of castor oil by gavage, and only those indicating diarrhea were chosen for the examination. The test animals were fasted overnight with free access to water and randomly distributed to six groups, each of six mice. Vehicles (distilled water containing 1% Tween-80) were only given to the animals of the control group (I) and loperamide (3 mg/kg; b.w., i.p.) as a standard antimotility drug to Group II (positive control). Other test groups (Group III, group IV, group V, and Group VI) were treated with oral doses of MEBS suspension at 50, 100, 200, and 400 (mg/kg b.w.), respectively. Mice of all groups received 0.5 mL of castor oil after one hour of administration of test samples. Then mice of all groups were kept on the enclosure’s transparent paper floor and facilitated with the same environmental conditions. All visible diarrheal symptoms were noted during the observational period, emphasizing the onset of diarrhea, weight, number of wet stools, and total fecal yields. Finally, the diarrheal inhibition (% inhibition of defecation) was determined following the formula [25] described below.
$$\%\;\mathrm{Inhibition}\;\mathrm{of}\;\mathrm{defecation}=\frac{\mathrm{Mean}\;\mathrm{defecation}\;\mathrm{of}\;\mathrm{control}}{\mathrm{Mean}\;\mathrm{defecation}\;\mathrm{of}\;\mathrm{the}\;\mathrm{test}\;\mathrm{sample}\;\mathrm{or}\;\mathrm{standard}\;\mathrm{drug}}\times 100$$

#### 2.9.2. Castor Oil-Induced Entero-Pooling

The intraluminal liquid accumulation technique mentioned by Robert et al. [26] was followed to accelerate this research. The mice were separated into six groups consisting of six mice in each group and fasted for 18 h. Distilled water containing 1% Tween-80 was given to the animals of the control group (I) as vehicles and loperamide (3 mg/kg; b.w., i.p.) to Group II (positive control) as a standard antimotility drug. Mice in the test groups (Group III, group IV, group V, and Group VI) were treated with oral doses of MEBS suspension at 50, 100, 200, and 400 (mg/kg b.w.), respectively. Then 0.5 mL of castor oil was given orally to the mice of each group after 1 h as the causative agent of diarrhea. After 2 h, an overdose of chloroform anesthesia was employed on the mice to sacrifice them, and the small intestinal tract at the point of pyloric sphincter and ileocecal junctions was ligated and deserted out. Then intestinal substances were collected by draining into a graduated tube, and the volume of the collected materials was estimated. The fluctuations of the complete and empty intestinal tracts were also determined, and the findings were contrasted to the average effect of the vehicle. Finally, the percent of intestinal secretions and weight of intestinal substances were calculated following the simultaneous equations [25].
$$\%\;\mathrm{of}\;\mathrm{inhibition}\;\mathrm{of}\;\mathrm{MVSIC}=\frac{\left(\mathrm{MVICC}-\mathrm{MVICT}\right)}{\;\mathrm{MVICC}}\times 100$$
$$\%\;\mathrm{of}\;\mathrm{inhibition}\;\mathrm{of}\;\mathrm{MVSIC}=\frac{\left(\mathrm{MVICC}-\mathrm{MVICT}\right)}{\;\mathrm{MVICC}}\times 100$$
where MVSIC is the mean volume of the small intestinal content, MVICT is the mean volume of the intestinal content of the test groups, and MVICC is the mean volume of the intestinal content of the control group.
$$\%\;\mathrm{of}\;\mathrm{inhibition}\;\mathrm{of}\;\mathrm{MWSIC}=\frac{\mathrm{MWICC}-\mathrm{MWICT}}{\mathrm{MWICC}}\times 100$$
where MWSIC is the mean weight of the small intestinal content, MWICC is the mean weight of the intestinal content of the control group & MWICT is the mean weight of the intestinal content of the test groups.

#### 2.9.3. Gastrointestinal Motility Test

This test was performed by implementing the method mentioned by Rudra et al. [27]. All experimental mice were fasted for 18 h and isolated into six groups comprising six mice in each. All mice were given castor oil orally to initiate diarrhea. Group I (negative control) received vehicles (distilled water containing 1% tween 80 orally), Group II (positive control) received standard drug (loperamide 3 mg/kg; b.w., i.p.), and group III, IV, V, and VI were treated with MEBS 50, 100, 200 and 400 (mg/kg; b.w, p.o), respectively, after an hour of the introduction of castor oil. Afterwards, the animals received 1 mL of Charcoal (10% charcoal suspension in 5% gum acacia) orally after an hour of oral administration of MEBS. Then overdose of chloroform was employed on animals to anaesthetize, prior to sacrificing them after 1 h. The distance moved by the charcoal meal in the intestinal tract from the pylorus to the caecum was estimated, and the total transit in the intestine was also designated. The peristaltic index and percentage of inhibition were estimated by following the executed formula [25].
$$\mathrm{Peristalsis}\;\mathrm{index}=\frac{\mathrm{Distance}\;\mathrm{travelled}\;\mathrm{by}\;\mathrm{charcoal}\;\mathrm{meal}\;}{\mathrm{Length}\;\mathrm{of}\;\mathrm{small}\;\mathrm{intestine}}\times 100$$
$$\%\;\mathrm{inhibition}=\frac{\left(\mathrm{MDc}-\mathrm{MDt}\;\right)}{\mathrm{MDc}\;}\times 100$$

### 2.10. In Vitro Bioassay

#### 2.10.1. Antibacterial Assay 

The bacterial growth inhibition analysis (antibacterial) of MEBS was carried out by the disc diffusion manner [24]. In this scheme, a definite amount of the test sample is dissolved in a particular solvent to give a solution of a given concentration (μg/mL). At that point, the sterile filter paper disc of 5 mm diameter is saturated with test samples of known amounts and dried. Such discs and standard antibiotic discs are set on the plate containing an appropriate nutrient agar medium seeded with the test organisms. The plates were kept at a low temperature (4 °C) for 24 h to permit maximum diffusion. The plates were then incubated at 37 °C for 18–24 h to permit the growth of organisms. However, a clear, distinct zone, termed “Zone of Inhibition,” will be achieved, inhibiting the growth of microorganisms, indicating possession of antibacterial action. The antibacterial action of the test sample was estimated by calculating the diameter of the zone of inhibition (mm), and each experiment was in triplicate. Amoxicillin 30 (mg/disc) was used as a reference standard.

#### 2.10.2. Antifungal Activity

The antifungal sensitivity test of the MEBS was also performed using the disc diffusion assay [28], similar to the antibacterial activity test. The difference was in the incubation time (72 h) and temperature (25 °C). The test organisms were cultured, and antifungal activity was performed in a medium of potato dextrose agar (PDA). The antifungal properties of the MEBS were estimated by converting the zone of inhibition (mm) of the 4 pathogens to the percentage zone of inhibition and subsequent counting total zone of inhibition (TZI) caused by the test groups in comparison to the standard drug fluconazole 20 (μg/mL). 

### 2.11. In Silico Screening

#### 2.11.1. Molecular Docking: Protein Preparation

Three dimensional crystal structure including M3 muscarinic acetylcholine receptor (PDB ID: 5ZHP) [29], human glutamate carboxypeptidase II (GCP II) (PDB ID: 4P4D6) [30], *E. coli* exonuclease I (PDB: 1XFF) [31], GPCR beta-arrestin (PDB: 6U1N) [32] and Cytochrome P450 14 alpha-sterol demethylase (PDB ID: 1EA1) [33] were picked from RCBS Protein Data Bank in PDB format. Thereupon, with the Discovery Studio 2020, all water and hetero-atom were extracted from the proteins. Proteins were arranged through combining and granting Gasteiger charge non-polar hydrogen. In addition, the least energy state of all proteins was confirmed by keeping the standard residues in AMBER f14sB mode. Seleno-methionine, co-methionine, Bromo-UMP, methyl-sulfonyl-dUMP to UMP, and methyl-sulfonyl-dCMP to CMP were also selected. The incomplete side chain was replaced using the Dunbrack rotamer library and processed for further analyses in the UCSF Chimera [34].

#### 2.11.2. Molecular Docking: Ligand Preparation 

Five compounds identified by high-resolution UPLC-QTOF–M.S analysis of *Bauhinia scandens* were selected based on their lowest molecular weight for the docking analysis. Chemical structure of the components, namely 6-hydroxykaempferol (PubChem CID: 5281638), galangin (PubChem CID: 5281616), iris-florentin (PubChem CID: 170569), luteolin (PubChem CID: 5280445), and retusine (PubChem CID: 5352005) were downloaded from the PubChem database [35]. The compounds were downloaded in 2DSDF. They have been converted to pdbqt format using PyRx tools [36] to check for the effective binding with the targeted proteins.

#### 2.11.3. Molecular Docking: Docking Analysis 

The protein-ligand linking framework of the selected protein-ligand complexes was determined by using PyRx Autodock Vina [37]. Initially, the proteins were formatted into a macromolecule, and a semi-flexible docking mechanism was introduced for the docking analysis. Using PyRx AutoDock software, the PDB format of the chemicals and proteins has been minimized and converted to PDBQT format. The protein rigidity and ligands flexibility were sustained during the analysis. The molecules and Ligand were given ten degrees of freedom. AutoDock defined the steps toward pdbqt molecules, box type, grid box formation, etc., in default formation. The grid box with an active position was formed at the center of the box. Finally, it was accelerated in BIOVIA Discovery Studio Visualizer 2020 [36] to determine the best binding approaches and docking positions.

#### 2.11.4. Ligand-Based Pharmacokinetics and Toxicity Measurement

SwissADME (http://www.swissadme.ch/) (22 January 2020) was used to determine the pharmacokinetic properties (ADME) of the three major compounds. Favourable drug-like properties of all selected compounds were evaluated considering Lipinski’s rule of five (M.W. not more than 500; H-bond donors ≤ 5; H-bond acceptors ≤ 10; Lipophilicity < 5), Human Intestinal Absorption probability, Human Oral Bioavailability probability, Plasma Protein Binding probability [38]. In addition, an online tool, admetSAR (http://lmmd.ecust.edu.cn/admetsar2) (22 January 2020) were employed to determine the toxicological properties (Carcinogenicity probability, AMES Mutagenesis probability, Acute Oral Toxicity) of all the compounds.

## 3. Statistical Analysis

The data of statistical analysis are presented as mean ± SEM. The information was controlled by using Graph pad prism version 5.0. Dunnett’s Multiple Comparison Test performed one Way Analysis of Variance (ANOVA) to determine statistical significance, and *p*-values of less than 0.05, 0.01, and 0.001 were considered. Whenever suitable, the dose–dependency features of the anti-diarrheal activity were observed applying linear regression analysis.

## 4. Results

### 4.1. Phytochemical Screening Result

The qualitative phytochemical analysis was implemented to confirm the existence of secondary plant metabolites. The assessment’s phytochemical screening results showed (Table 1) that MEBS highly yields Tannins, Flavonoids, Glycosides, Quinones, Sterols, and Saponins. Besides, Polyphenols, and Terpenoids were also moderately present; Steroids were found in small quantities.

### 4.2. Chemical Analysis by the UPLC-QTOF-M.S. Method

In 20 min retention time, the methanol extract of *B. scandens* exhibited a total of 218 flavonoids (Appendix A) eluted between 0–17 min. 

### 4.3. Oral Acute Toxicity

Animals were not dead or morbid for the 14 days following single oral treatment. Moreover, typical morphology, including fur, eyes, and nose, appeared normal. Besides morphology, abnormal physiological features such as diarrhea, lethargy, salivation, tremors, hallucinations, and/or irregular behavior such as self-mutation, regressive actions, etc., were absent. Behavioral features such as gait and attitude, sensory stimuli, and controlling reactivity were also natural. Therefore, the weight of the control and the treatment groups was stable. Food and water supply to the mice were routinely controlled. These features prove that a single body-weight dose of 2000 mg/kg of the MEBS was stable. Hence, MEBS doses for the anti-diarrheal activity were chosen at a dose of 50, 100, 200, and 400 mg/kg.

### 4.4. Anti-Diarrheal Assay

#### 4.4.1. Castor Oil-Induced Diarrhea

Significant (*p* < 0.05, *p* < 0.01, *p* < 0.001) dose-dependent delay in the onset of diarrhea has been observed after administration of MEBS (50, 100, 200 and 400 mg/kg). Meanwhile, the number and weight of wet feces and the total number and weight of all feces were also significantly reduced. The castor oil-induced diarrhea has been summarized and shown in Table 2. However, the data have shown the percentage of diarrhea (number of wet feces) inhibitions in a dose-dependent manner and were 22.31%, 38.79%, 58.15%, and 80.64% at doses of 50, 100, 200, and 400 (mg/kg) subsequently, where the reference drug (Loperamide-3 mg/kg) inhibited 88.08% of diarrhea.

#### 4.4.2. Castor Oil-Induced Entero-Pooling

The volume and weight of intestinal substances were decreased significantly (*p* < 0.05, *p* < 0.01, *p* < 0.001) as a consequence of the treatment of MEBS, as contrasted individually with control (Table 3). In MWSIC, the percentage of inhibitions were found to be 16.66%, 38.33%, 55.00%, 71.66% at MEBS 50, 100, 200, 400 (mg/kg), respectively, and the percentage inhibition of MVSIC were found to be 14.00%, 44.00%, 60.00%, and 68.00% for MEBS 50, 100, 200, and 400 (mg/kg), respectively. On the contrary, the inhibition rate of Loperamide (3 mg/kg) was 71.67% in MWSIC and 72.91% in MVSIC, respectively. 

#### 4.4.3. Charcoal Induced Intestinal Transit in Mice

The charcoal induced gastrointestinal transit in mice was significantly reduced in a dose-dependent manner at 11.11%, 33.01%, 50.08%, and 79.10%, at the doses of MEBS 50, 100, 200, and 400 (mg/kg), respectively (*p* < 0.05, *p* < 0.01, *p* < 0.001). The antimotility effects of the maximum dose of the extracts were similar to the standard drug loperamide (3 mg/kg), which attained 78.83% of transit inhibition in mice. Besides, the total length of the small intestine, distance moved by the Charcoal in the small intestine, inhibition of gastrointestinal transit of Charcoal by MEBS, and peristaltic index (%) were also calculated. All findings have been listed in Table 4.

### 4.5. Antibacterial Assay 

Bacterial growth inhibition activity of MEBS against eight pathogenic bacteria was tested, and the zones of inhibition (mm) were measured. The results showed dose-dependent inhibition of bacterial growth in the disc and have been illustrated in Table 5. With the increase of concentration, MEBS revealed better action against almost all the treated bacteria except *Pseudomonas aeruginosa*. From the tabular analysis of MEBS against eight pathogenic bacteria, it can be suggested that the MEBS cannot act against *Pseudomonas aeruginosa* and significantly inhibited the growth of *Staphylococcus aureus, Lactobacillus casei, Bacillus azotoformans, Corynebacterium species, Bacillus cereus,*
*Salmonella typhi*, and *Escherichia coli* strains. The growth of *Bacillus azotoformans, Staphylococcus aureus,* and *Salmonella typhi* was potentially inhibited by MEBS 500 (µg/disc). The maximum zone of inhibition (26 mm) was attained for MEBS 500 (µg/disc) against *Staphylococcus aureus,* and standard drug amoxicillin revealed a 34 mm zone of inhibition against *Bacillus azotoformans*. In addition, the maximum zone of inhibition (25 mm) was achieved against *Salmonella typhi* at 500 (µg/disc) among all the gram-negative bacteria, where Amoxicillin revealed a 20 mm zone of inhibition against *Salmonella typhi* at 30 (µg/disc). 

### 4.6. Antifungal Screening

From the examination, it can be reported that MEBS possesses moderate dose-dependent inhibition of growth against some fungal strains. The maximum zone of inhibition was obtained at 20 mm for *Candida albicans* and *Blastomyces dermatitidis* at MEBS 300 (µg/disc) where fluconazole 20 (µg/disc) yields a 36 and 30 mm zone of inhibition against *Candida albicans* and *Blastomyces dermatitidis*, respectively. The study also revealed that MEBS 300 (µg/disc) inhibited 16 mm of the growth of *Trichoderma* spp. Besides, MEBS 100 (µg/disc) showed the inhibition of 12 mm and 10 mm of growth inhibition and 15 mm and 13 mm of growth inhibition against *Candida albicans*, *Blastomyces dermatitidis*, *Trichoderma* spp., respectively. Besides, MEBS did not show any sensitivity against *Cryptococcus neoformans*. The judgment of the antifungal activity of the plant extract is illustrated in Table 6.

### 4.7. Docking Analysis for the Anti-Diarrheal, Antibacterial and Antifungal Studies

In the anti-diarrheal docking study, M3 muscarinic acetylcholine receptor (PDB ID: 5ZHP) and human glutamate carboxypeptidase II (PDB ID: 4P4D) were docked against five selected compounds that were found from the UPLC-QTOF-M.S. analysis of MEBS. The binding affinity was measured based on docking score, and regarding this score, Galangin (−6.2 kcal/mol) and 6-Hydroxykaempferol (−7.9 kcal/mol) illustrated the best binding affinity to the M3 muscarinic acetylcholine receptor and human glutamate carboxypeptidase II receptor, respectively. Again, Luteolin and Irisflorentin showed the most negligible binding affinity to the receptors subsequently. 6-Hydroxykaempferol binds to the human glutamate carboxypeptidase II by a series of amino acids: conventional hydrogen bond (tyr234, tyr552), carbon-hydrogen bond (gln254, glu457), pi-cation (zn816), pi-pi stacked (leu384), pi-alkyl (tyr205). Galangin also binds to the M3 muscarinic acetylcholine receptor by a pi-pi stacked (phe554), pi-alkyl (lys555, leu558, arg551, ala544) (Figure 1, Table 7). 

Glutaminase domain (PDB ID: 1XFF), GPCR-Beta arrestin (PDB ID: 6U1N) was docked against selected compounds to assess the antibacterial binding affinity of the compounds. Luteolin showed maximum binding affinity (−6.1 kcal/mol) and Retusine exhibited the most miniature binding (−4.40 kcal/mol) affinity with the Glutaminase domain receptor. The docking score was formed as follows: Luteolin > 6-Hydroxykaempferol > Amoxicillin > Galangin > Retusine. Retusine (−9.1 kcal/mol) and Irisflorentin (−5.7 kcal/mol) explored the highest and lowest binding affinity to the GPCR-Beta arrestin receptor. Retusine binds to the GPCR-Beta arrestin receptor throughout the amino acid residues: conventional hydrogen bond (arg282, lys292), carbon-hydrogen bond (gly137), and pi-pi T shaped (his295), alkyl (pro131), pi-alkyl (pro133) (Figure 1, Table 7).

The antifungal docking study was also implemented where 6-Hydroxykaempferol disclosed the highest binding affinity to the Cytochrome P450 14 alpha-sterol demethylase (1EA1). 6-Hydroxykaempferol binds to the Cytochrome P450 14 alpha-sterol demethylase via conventional hydrogen bond (gln72, arg326), carbon-hydrogen bond (phe63, ala389, his292) and pi-sigma (leu321, thr260), alkyl (pro320), pi-alkyl (leu324, cys394) (Figure 1, Table 7). 

### 4.8. Compound Based Pharmacokinetics and Toxicity Property Analysis

The pharmacokinetic compounds described by Lipinski’s five rules can easily be obtained from the SwissADME online site [38]. The ADME characteristics (absorption, distribution, metabolism, elimination) of compounds are exhibited in Table 8. The cell permeation, bioavailability, and metabolism were defined for all selected features. The predicted properties of compounds were kept within limits.

## 5. Discussion

Since ancient history, plant derivatives and phytochemicals have been utilized as folk medicine. Therefore, scientific study has explored the actual uses of phytochemicals in the pharmaceuticals industry [39,40,41]. Some phytochemicals are not involved in the natural physiology of the plant [42]. Extraction, isolation, and purification processes of such phytochemicals have been established as advancements in the pharmaceutical industry. Conversely, one of the big hurdles during the extraction procedures is that co-extraction interferences with the target compounds [43]. Recently, a variety of research work has suggested UPLC-QTOF-M.S for quality control assessment of several phytomedicines [44]. UPLC-QTOF-M.S. analysis is crucial due to sensitivity and its specificity and enhanced resolution support by UHPLC systems. The system also corroborates retrospective data analyses of the provided sample. However, occasionally there may be a presence of multiple phytoconstituents during the analysis, but this might be a result of the presence of a variety of multiple samples during the analysis. 

The discovery of effective medicines from plant sources has diminished the consequences of infectious illnesses and improved individual satisfaction. Moreover, some ingredients might be utilized in better ways rather than conventional medications to improve health [45]. Plant-originated medicine might be an appealing, regular and affordable source in antimicrobial treatment, particularly for less developed societies [46]. 

Diarrhea results from the rapid development of fecal stuff through the large intestine [47]. Castor oil was used in all methods for inducing diarrhea because castor oil introduces diarrhea due to ricinoleic acid, an active metabolite liberated in the small intestine by lipase action [48]. Intestinal lipases produce ricinoleic acid responsible for the activity of castor oil. Although castor oil is extensively used in conventional and folk medicine, the molecular mechanism of ricinoleic acid is still unclear. Ricinoleic acid stimulates the Prostanoid EP3 receptor, which substantially initiates the pharmacological effects of castor oil [49]. The laxative effect of ricinoleic acid and uterine contraction is absent in mice, with no EP3 receptors. While the EP3 receptor generation in intestinal epithelial cells was deleted conditionally, it did not trigger diarrhea attributed to castor oil. Mice without EP3 receptors in smooth muscle cells had an unusual response to this drug. Therefore, EP3 prostanoid receptors in the intestinal and uterine cells activate the castor oil, which subsequently triggers the metabolic pathways of ricinoleic acid [50]. Such description of cellular and molecular pathways displays the pharmacological rules of castor oil known so far, and demonstrate the relevance to the laxative effects of the EP3 receptor [51]. Castor oil-induced diarrhea has been used to evaluate the onset of diarrhea and the number and frequency of wet feces. In our investigation, the fecal time was delayed, the weight of the wet feces was retarded, and the frequency of wet feces was reduced by MEBS beyond that of the castor oil-induced diarrhea produced in the mice model. The dose-dependent potentiality of the MEBS in terms of percentage of inhibition rate of feces was primarily found in 200 mg/kg and 400 mg/kg upon contrast with the control. The effect of MEBS 400 mg/kg is likely to the Loperamide (3 mg/kg), which is used as a standard positive control. Additionally, the retardation of onset of diarrhea, weight of wet feces, and frequency of diarrhea inhibited by administering MEBS indicates the existence of the anti-diarrheal potentiality of MEBS.

The entero-pooling model evaluated the secretory constituents of diarrheal disorder. This study showed the significant efficacy of all tested doses of MEBS extract in MWSIC and MVSIC compared to the positive control. In the present study, it has been distinguished that castor oil is liable to diarrheal activity because it contains nitric oxide. This diarrheal effectiveness includes reducing general liquid misappropriation by obstruction of intestinal Na⁺, K⁺ ATPase activity mediated by dynamic secretion of adenylate cyclase or mucosal cAMP [52]. Castor oil possesses ricinoleic acid, an active metabolite capable of triggering the nitric oxide pathway and, substantially, nitric oxide (NO) provokes gut secretion [53]. MEBS *(p* < 0.05, *p* < 0.01, *p* < 0.001) lessens the secretory effect significantly, which was propagated by nitric oxide as well as ricinoleic acid. Therefore, It can be presumed that the presence of flavonoids implicated in attenuation of NO synthesis [54] and MEBS contains these types of substances, which presume to act against NO implicated defecation. Regarding declaration [55], it can be reported that the antisecretory effects of MEBS may be observed due to the presence of tannin and flavonoids.

Most anti-diarrheal agents reduce gastrointestinal motility; hence, the charcoal meal method was chosen during the analysis to pursue the dislocation of the gastrointestinal materials in the presence of diarrheal and anti-diarrheal agents [56]. Activated Charcoal has been an essential tool for assessing the impact of laxatives and using them as a marker in the gastrointestinal transit model for more than 60 years [57]. This strategy is a pointer to determine the movement of activated Charcoal as a marker in the small intestine [58]. This principle was employed to evaluate the dose-dependent efficacy of MEBS in order to reduce the conduction of the charcoal marker. The peristaltic index and the traveling distance of the charcoal marker were least in the presence of 200 mg/kg and 400 mg/kg (b.w.) MEBS contrasted with the control. This result ensures that the MEBS extracts evenly act on the entire intestinal tract. Therefore, retardation in the motility of intestinal muscles promotes substances to stay in the intestinal tract for a long time [59]. This permits better water absorption from the gut. Such medications restrain intestinal transit and promote relief of diarrhea in pathophysiological states [60]. However, the antispasmodic properties of the test extract eventually decrease the intestinal propulsive development in the gastrointestinal transit model [61]. Secondary metabolites, for example flavonoids and tannins, are accounted to have anti-diarrheal movement because of their capacity to restrain intestinal motility. Subsequently, the synergistic inhibitory impact of saponins and tannins influences the overall anti-diarrheal impact of the test extract on the Charcoal-induced gastrointestinal transit test [62]. Therefore, it is also claimed that numerous plants show anti-diarrheal activity with supposed antibacterial potentiality [63]. MEBS demonstrated high susceptibility against some pathogenic bacteria, and it is also sensitive to some pathogenic fungi. It also contains tannins, flavonoids, alkaloids, saponins, and phenolic compounds, which might be the significant reasons for the anti-diarrheal action, aside from the fact that its antisecretory antimotility and gastrointestinal transit hindrance impacts were observed in this investigation. These remarkable antimicrobial properties also reinforce the belief that MEBS can be a powerful candidate for treating various etiologies of diarrhea, including infectious segments.

Our study also focused on the antimicrobial effect of MEBS against some positive and negative pathogenic bacteria. It was revealed from the investigation that MEBS showed high compassion to wide-spectrum bacterial strains. Besides, the phytochemical evaluation of the current study depicted the existence of tannins, flavonoids, glycosides, quinones, sterols, saponins, polyphenols, terpenoids, steroids, and amino acids. Quinones are omnipresent compounds found in many natural products, particularly herbs, fungi and bacteria. This activity may be attributed to the polyphenolic content. In addition, a recent study showed that a wide range of microorganisms was hindered by polyphenolic compounds [64]. The antimicrobial behaviour of phenol and tannin have already been established [65,66]. Tannins possessed antimicrobial efficacy against microbes [67]. On average, Gram-positive bacteria yielded the most inhibition compared to Gram-negative bacteria. Many researchers reported that Gram-negative pathogenic bacteria are more resistant to organic extracts of plants, as Gram-negative bacteria contain a hydrophilic cell wall, mainly consisting of a lipopolysaccharide that prevents the penetration of hydrophobic elements and prevents organic extract from acting at the target cell surface [68].

A study shows the possible extent of lipophilicity in extracts that have increased the activity of plant extract as a standard drug, due to interactions and arrangements of lipophilicity with the membrane components [69]. According to the above, Gram-positive bacteria are more sensitive because of having only an external peptidoglycan layer that is very seldom adequate protection against permeability [70]. Again, MEBS also formed moderate antifungal effects against fungi. The zone of inhibition of MEBS has been compared with the standard doses of drugs applied to conduct the antifungal test. The antifungal study of MEBS revealed that *Candida albicans*, *Blastomyces dermatitidis*, and *Trichoderma* spp. are sensitive to the MEBS and *Cryptococcus neoformans* is resistant to the test extract. Besides, it has been claimed that flavonoids containing plants show better susceptibility against several pathogenic fungi [71]. The pathways supposed to be accountable for such phytochemicals against pathogens have been varied and dependent mainly on the enzyme inhibition of these substances by the oxidation of elements, and act as a source of reliable free radicals, contributing to the protein inactivation functional loss of pathogens. These are capable of compellation with extracellular, soluble proteins and the complex bacterial cells terminating microbial membranes. Some can interpret DNA, ion channel formation in the microbial membrane, and competitive retardation in the host of polysaccharide receptors in microbial proteins [72]. Therefore, several Gram-positive and Gram-negative bacteria and some fungi showed susceptibility against MEBS. Hence, it can be inferred that MEBS can be the source of antimicrobial agents.

Molecular docking is a modern and helpful technique to predict the binding efficacy of ligands with the target proteins and helps achieve better insights into the biological activity of the phytoconstituents. In addition, it can facilitate a better understanding of the binding efficacy of possible molecular mechanisms within various enzymatic pockets [73]. Henceforth, five representative components of MEBS were docked against four target receptors, and the computational findings were correlated with experimental results. In our experiment, the observed biological activities are anti-diarrheal, antibacterial, and antifungal, and the four targets we have selected were M3 muscarinic acetylcholine receptor (PDB ID: 5ZHP), human glutamate carboxypeptidase II (PDB ID: 4P4D), glucosamine 6-phosphate synthase (PDB ID: 1XFF), GPCR-Beta arrestin (PDB ID: 6U1N) and Cytochrome P450 14 alpha-sterol demethylase (CYP51, PDB ID: 1EA1). Molecular docking studies with the Glutaminase domain (PDB ID: 1XFF), GPCR-Beta arrestin (PDB ID: 6U1N) revealed the antibacterial activity of our identified compounds of MEBS. Among the five compounds, four compounds, excluding iris-florentin, exhibited binding affinity with the active sites of the glutaminase domain and GPCR-Beta arrestin receptor. The antifungal molecular docking study was carried out using Cytochrome P450 14 alpha-sterol demethylase (PDB ID: 1EA1) as our target protein. The visualization and results of docking analysis indicate that the selected compounds interact with targeted enzymes by a series of chemical bonds. We selected Amoxicillin as the standard drug and compared it to the binding affinities of the selected compound retrained from the chromatography (UPLC-QTOF–M.S.) of the methanol extract of the *B. scandens* stems. In both cases, the binding affinity was more significant than our standard Amoxicillin. So, the selected compounds of MEBS may exhibit antibacterial activity through interaction with these target proteins. We can conclude that the identified compounds may be a phytochemical or flavonoid source that possesses the anti-diarrheal, antibacterial and antifungal properties of MEBS.

## 6. Conclusions

The study aimed to validate the application of *Bauhinia scandens* L. stems as anti-diarrheal substance in conventional folk medicine. In our investigation, it is transparent that MEBS can be another wellspring of antibacterial and antifungal agents against several pathogenic strains. It is additionally assumed that the antimicrobial effect of MEBS may be associated with its chemical composition, which also provokes anti-diarrheal activity. Therefore, further examination is required to identify other dynamic constituents in charge of anti-diarrheal, antimicrobial, and antifungal potential.

## Figures and Tables

**Figure 1 nutrients-14-00265-f001:**
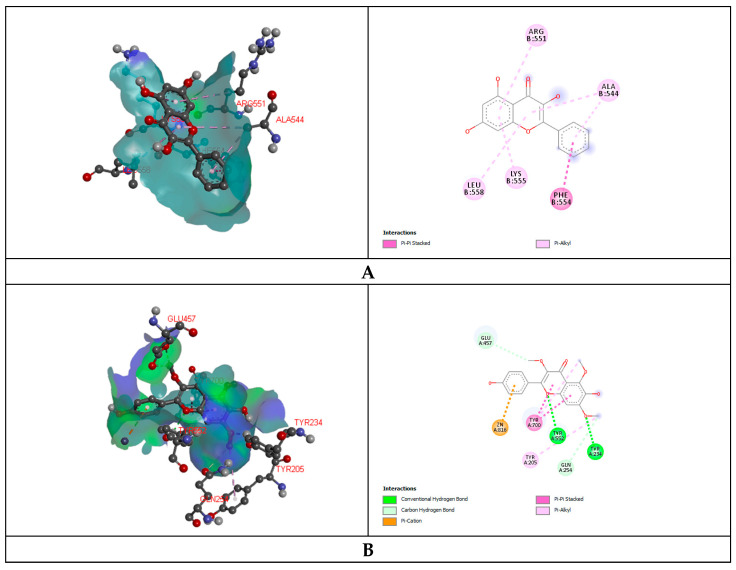
Representation (2D and 3D) docking results of best ranked pose of key interactions in the binding pocket of M3 muscarinic acetylcholine receptor (PDB ID: 5ZHP) with Galangin (**A**), human glutamate carboxypeptidase II (GCPII) with 6-Hydroxykaempferol (**B**), *E.coli* exonuclease I with Luteolin (**C**), GPCR beta arrestine with retusine (**D**), Cytochrome P450 14 alpha-sterol demethylase with 6-Hydroxykaempferol (**E**) respectively.

**Table 1 nutrients-14-00265-t001:** Phytochemical screening of the methanol extract of *Bauhinia scandens* stems.

Group Test	Observation
Tannins	+
Glycosides	+
Quinones	+
Sterols	+
Saponins	+
polyphenols	+
Terpinoids	+
Steroids	+
Flavonoids	+
Protein	-
Starch	-
Phlobatonin	-
Xanthoprotein	-
Anthraquinones	-
Coumarins	-
Cholesterols	-
Anthracins	-
Carbohydrate	-
Emodins	-
Leuco-anthocyanin	-
Oxalates	-
Resins	-
Amino Acid	-
Triterpines	-
Vitamin C	-

(+) = Present, (-) = absent.

**Table 2 nutrients-14-00265-t002:** Effect of the test samples on castor oil induced diarrhea in mice.

Treatment	Dose, Route (p.o)	Onset of Diarrhea (min)	Average Number of Wet Feces	Average Number of Total Feces	Average Weight of Wet Feces (g)	Average Weight of Total Feces (g)	% of Inhibition of Wet Feces (Defecation)
Group-I	1% tween 80–10 mL/kg	77.16 ± 1.83	11.16 ± 0.79	13.67 ± 0.80	0.37 ± 0.02	0.42 ± 0.02	-
Group-II	Loperamide-3 mg/kg (i.p)	195.5 ± 8.91 ***	1.33 ± 0.21 ***	2.16 ± 0.30 ***	0.04 ± 0.007 ***	0.07 ± 0.1 ***	88.08
Group-III	MEBS-50 mg/kg	84.16 ± 2.82	8.67 ± 0.49 *	11.33 ± 0.80	0.24 ± 0.01 *	0.36 ± 0.03 *	22.31
Group-IV	MEBS-100 mg/kg	107 ± 4.02 **	6.83 ± 0.47 **	8.33 ± 0.67 *	0.21 ± 0.02 **	0.27 ± 0.02 **	38.79
Group-V	MEBS-200 mg/kg	132.33 ± 5.38 ***	4.67 ± 0.49 ***	6.83 ± 0.60 ***	0.16 ± 0.01 ***	0.21 ± 0.01 ***	58.15
Group-VI	MEBS-400 mg/kg	185.83 ± 6.28 ***	2.16 ± 0.30 ***	3.83 ± 0.40 ***	0.08 ± 0.01 ***	0.13 ± 0.01 ***	80.64

The data were presented as Mean ± SEM (*n* = 6); One Way Analysis of Variance (ANOVA) followed by Dunnett’s Multiple Comparison Test; * *p* < 0.05, ** *p* < 0.01, *** *p* < 0.001 were considered significant compared to the control sample. MEBS = Methanol extract of *Bauhinia scandens* L. stems.

**Table 3 nutrients-14-00265-t003:** Impact of the test samples on castor oil incited intraluminal fluid accumulation in mice.

Treatment	Dose, Route (p.o)	MWSIC (g)	% of Inhibition	MVSIC (mL)	% of Inhibition
Group-I	1% tween 80–10 mL/kg	0.60 ± 0.02	-	0.50 ± 0.02	-
Group-II	Loperamide-3 mg/kg (i.p)	0.17 ± 0.09 ***	71.66	0.14 ± 0.01 ***	72.00
Group-III	MEBS-50 mg/kg	0.50 ± 0.02	16.66	0.43 ± 0.01	14.00
Group-IV	MEBS-100 mg/kg	0.37 ± 0.01 **	38.33	0.28 ± 0.01 **	44.00
Group-V	MEBS-200 mg/kg	0.27 ± 0.01 ***	55.00	0.20 ± 0.01 ***	60.00
Group-VI	MEBS-400 mg/kg	0.17 ± 0.02 ***	71.66	0.16 ± 0.01 ***	68.00

The data are presented as Mean ± SEM (*n* = 6); One Way Analysis of Variance (ANOVA) followed by Dunnett’s Multiple Comparison Test; ** *p* < 0.01, *** *p* < 0.001 were considered significant compared to the control sample. MEBS = Methanol extract of *Bauhinia scandens* L. stems.

**Table 4 nutrients-14-00265-t004:** Impact of the test samples on intestinal transit in mice.

Treatment	Dose, Route (p.o)	Total Intestinal Length(cm)	Distance Moved by the Charcoal Meal (cm)	Peristaltic Index (%)	% of Inhibition
Group-I	1% tween 80–10 mL/kg	53.42 ± 2.02	38.87 ± 1.80	72.76	-
Group-II	Loperamide-3 mg/kg (i.p)	49.67 ± 0.67	7.65 ± 0.76 ***	15.40	78.83
Group-III	MEBS-50 mg/kg	50.83 ± 1.16	32.07 ± 1.31	63.09	11.11
Group-IV	MEBS-100 mg/kg	50.00 ± 1.77	24.37 ± 1.41 **	48.74	33.01
Group-V	MEBS-200 mg/kg	51.67 ± 1.49	18.77 ± 1.08 ***	36.32	50.08
Group-VI	MEBS-400 mg/kg	50.17 ± 0.67	7.63 ± 0.76 ***	15.20	79.10

The data were presented as Mean ± SEM (*n* = 6); One Way Analysis of Variance (ANOVA) followed by Dunnett’s Multiple Comparison Test; ** *p* < 0.01, *** *p* < 0.001 were considered significant compared to the control sample. MEBS = Methanol extract of *Bauhinia scandens* L. stems.

**Table 5 nutrients-14-00265-t005:** Antibacterial activities of the test samples against several food poisonous bacteria (gram positive and gram negative).

Test Organisms	Diameter of the Zone of Inhibition (mm)			
	MEBS (100 μg/disc)	MEBS (300 μg/disc)	MEBS (500 μg/disc)	Amoxicillin (30 μg/disc)
*Staphylococcus aureus*	17	21	26	32
*Lactobacilllus casei*	19	20	25	36
*Bacillus azotoformans*	15	19	22	34
*Corynebacterium species*	14	19	24	35
*Bacillus cereus*	15	20	23	33
*Salmonella typhi*	16	22	25	20
*Escherichia coli*	15	19	22	34
*Pseudomonas aeruginosa*	-	-	-	34

**Table 6 nutrients-14-00265-t006:** Antifungal profile of test samples against certain fungi.

Test Organisms	Diameter of Zone of Inhibition (mm)			
	MEBS-100 µg/disc	MEBS-300 µg/disc	MEBS-500 µg/disc	Fluconazole-20 µg/disc
*Candida albicans*	11	15	20	32
*Cryptococcus neoformans*	20	24	28	30
*Blastomyces dermatitidis*	12	20	24	36
*Trichoderma* spp.	13	16	22	34

**Table 7 nutrients-14-00265-t007:** Docking score of 6-Hydroxykaempferol, Galangin, Irisflorentin, Luteolin, Retusine, and Loperamide with M3 muscarinic acetylcholine receptor (PDB ID: 5ZHP), human glutamate carboxypeptidase II (PDB ID: 4P4D), Glutaminase domain (PDB ID: 1XFF), GPCR-Beta arrestin (PDB ID: 6U1N), and Cytochrome P450 14 alpha-sterol demethylase (1EA1) in kcal/mol.

Compounds	5ZHP	4P4D	1XFF	6U1N	1EA1
	Docking Score	Docking Score	Docking Score	Docking Score	Docking Score
6-Hydroxykaempferol	−6.0	**−7.9**	−5.8	−7.9	−8.8
Galangin	**−6.2**	−7.1	−4.7	−7.8	−7.7
Irisflorentin	-	−5.0	-	−5.7	−4.5
Luteolin	−5.9	−7.6	**−6.1**	−8.7	**−8.7**
Retusine	−6.0	−5.8	−4.4	**−9.1**	−6.3
Loperamide/Amoxicillin/Fluconazole	−7.1	−5.1	−5.8	−5.7	−5.7

**Table 8 nutrients-14-00265-t008:** Absorption, digestion, metabolism, excretion, and toxicological (ADME/T) properties of the compounds for good oral bioavailability.

Molecules	PubChem CID	MW (g/mol)	HBD	HBA	LogP (o/w)	HIA	HOB	PPB (100%)	CAR (Binary)	AM	AOT (kg/moL)
6-Hydroxykaempferol	5281638	302.23	5	7	1.99	0.9833	0.5571	1.006	1.0000	0.5400	1.545
Galangin	5281616	270.24	3	5	2.58	0.9881	0.5000	1.102	1.0000	0.7900	2.046
Irisflorentin	170569	386.4	0	8	3.22	0.9833	0.6286	0.852	0.9602	0.5000	2.708
Luteolin	5280445	286.24	4	6	2.28	0.9833	0.5714	1.043	1.0000	0.5100	2.525
Retusine	5352005	358.3	1	7	3.20	0.9889	0.5286	1.202	1.000	0.5400	2.086

PID = PubChem ID, MW = Molecular Weight (acceptance range: <500), HBD = Hydrogen Bond Donor (acceptance range: ≤5), HBA = Hydrogen Bond Acceptor: (acceptance range: ≤10), LogP = High Lipophilicity (acceptance range: <5), HIA = Human Intestinal Absorption probability, HOB = Human Oral Bioavailability probability, PPB = Plasma Protein Binding probability, CAR = Carcinogenicity probability, AM = AMES Mutagenesis probability, AOT = Acute Oral Toxicity.

## Data Availability

All the data have been inserted in the manuscript and Appendix A.

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
