# Peer review of "Chemical, Pharmacological and Computerized Molecular Analysis of Stem’s Extracts of *Bauhinia scandens* L. Provide Insights into the Management of Diarrheal and Microbial Infections"

_nutrients, 2022, doi:10.3390/nu14020265_

Round 1

Reviewer 1 Report

The work seeks to establish the capacity of an extract of Bauhinia scandens L. to treat diarrhea and to behave as an antibacterial and antifungal product. In addition, it aims to identify the compounds and potential mechanisms involved in this process. It is an interesting work; it has a lot of information and results; however, some improvements should be considered to be published.

Specifically, consider rewriting the introduction to appreciate a common thread that presents the problem or opportunity and the reason for making each of the determinations assessed. For example, after talking about diarrhea, they talk about the microorganisms that produce it (bacteria) and the potential use of natural compounds. Still, there is no logical continuity with the topic of antifungal products. Improve this part so that the writing does not appear to be a compilation of things made but rather writing with a single purpose.

In other aspects, check the English of the whole document, check the font type/font size across the entire document (there are different sentences-paragraphs); complete in methodology brand, model, the origin of some equipment (for example, line 102, high-capacity grinder); complete in the methodology section information referring to m.o. used (for example, indicate if they are m.o. wild, acquired, encoding, etc.). 

Rewrite point 2.6 (page 3, from line 126). It is written as if the mentioned compounds had already been detected (affirms their detection). Here it must be explained how they are seen or which ones will be detected. Better explain point 2.8, since it is indicated that the study will be carried out at 2000 mg/kg, but the analysis at different concentrations is mentioned. Explain that such concentrations are for further analyses.

In line 148, it is the chromatographic column (include "column" word).

Better explain what the% inhibition of defecation corresponds to. It is calculated with mass of defecation or number of defecations.

Point 2.9.2 could be reduced considering that the groups of animals and their treatments are the same, and what makes the difference are the analyzes to be carried out.

On line 197 use empty instead of void. Empty” implies that something was at one time filled. “Void” is that which is neither full.

In line 235: overnight = 24 h ??? 

In table 1, specify what corresponds to moderately present, and moderately present.

In the following tables (tables 2,3,and 4) indicate that group 1 is the control.

check parenthesis online 431-432 . 

Author Response

Dear editor

Thank you very much for your valuable comments to improve the quality of our manuscript. I hope you will consider the corrections regarding our research article entitled “Chemical, pharmacological and in silico studies of stem's extracts of Bauhinia scandens L. provide insights into the management of diarrheal and microbial infections” (Manuscript number: nutrients-1520213) feasibly. We have made corrections following the suggestions of the honorable reviewers. Finally, the corrections were marked with the highlighter. Please see the attachment.

Sincerely yours

Mohammad Aktar Sayeed

Department of Pharmacy

Faculty of Science and Engineering

International Islamic University Chittagong, Chattagram 4318, Bangladesh.

Email: sayeed_ustc@yahoo.com

Reviewer 2 Report

This paper extends the knowledge about Bauhinia scandens L. In a recent article, several authors of the present manuscript demonstrated the analgesic, anti-inflammatory and antipyretic potential of Bauhinia scandens, a subtropical region plant belonging to Fabaceae family. Here, a methanol stem extract showed potential in treating diarrhea and bacterial and fungal infections.

The study was well performed but I suggest several revisions. Following are my suggestions and questions:

- the English should be checked in the whole document preferably by an English speaking person (past tense should be used instead of past participle if possible; some words were misused, such as “However” in the last sentence in Abstract) 

- the Abstract section should be rephrased (words are repeating; “analyses” if plural; “bacterial and fungal strains” instead of “bacteria and fungi strains”)

- was a plant identification done or a plant voucher deposited?

- what was the ratio solvent:plant matrix?

- please provide the source for all chemicals used even if they were from your lab

- “Bauhinia scandensinstead of “Bauhinia Scandens” throughout the document

- “UPLC-QTOF–MS” instead of “UPLC-QTOF–M.S.” throughout

- “Table 1 (2, 3, ...)” instead of “table 1 (2, 3, ...)

- “14 α-sterol demethylase” instead of “14 alpha-sterol demethylase” 

- please define all abbreviations when first used

- some nouns do not need to be capitalized, such as luteolin, irisflorentin, galangin

- line 109 - “no one” meaning more than one?

- line 111 - “obtained from leaves”. The matrix is stem!

- line 124 - the mice were 8 weeks old or 7-8 weeks old? (please see next!)

- lines 161-166 - please develop this paragraph. Acute oral toxicity should be done with “each animal, at the commencement of its dosing, between 8 and 12 weeks old... with the mouse, food but not water should be withheld for 3-4 hours... After the substance has been administered, food may be withheld for a further 1-2 hours in mice.” (citation: OECD (2002), Test No. 420: Acute Oral Toxicity - Fixed Dose Procedure, OECD Guidelines for the Testing of Chemicals, Section 4, OECD Publishing, Paris, https://doi.org/10.1787/9789264070943-en)

- lines 183, 200, 222 - sentences need citations

- line 270 - PubChem database (citation) 

- line 275 - you should cite: Trott O, Olson AJ. AutoDock Vina: improving the speed and accuracy of docking with a new scoring function, efficient optimization, and multithreading. J Comput Chem. 2010 Jan 30;31(2):455-61. doi: 10.1002/jcc.21334

- line 384 - flucloxacillin is an antibiotic; did you mean fluconazole? 

- line 430 - add reference: Lipinski CA, Lombardo F, Dominy BW, Feeney PJ. Experimental and computational approaches to estimate solubility and permeability in drug discovery and development settings. Adv Drug Deliv Rev. 2001 Mar 1;46(1-3):3-26. doi: 10.1016/s0169-409x(00)00129-0

- lines 532-533 - “previous studies” instead of “a previous study” and add the following recent article that emphasizes your statement (doi: 10.3390/molecules25092187)

- in the Discussion section some ideas were repeated

Author Response

Dear Editor,

Thank you very much for your valuable comments to improve the quality of our manuscript. I hope you will consider the corrections regarding our research article entitled “Chemical, pharmacological and in silico studies of stem's extracts of Bauhinia scandens L. provide insights into the management of diarrheal and microbial infections” (Manuscript number: nutrients-1520213) feasibly. We have made corrections following the suggestions of the honorable reviewer. Finally, the corrections were marked with the highlighter. Please see the attachment.

Sincerely yours

Mohammad Aktar Sayeed

Department of Pharmacy

Faculty of Science and Engineering

International Islamic University Chittagong, Chattagram 4318, Bangladesh.

Email: sayeed_ustc@yahoo.com

Round 2

Reviewer 2 Report

There are still several suggestions not addressed:

- Line 19 - “was aimed” instead of “ was aim”

- Line 20 - “preliminary and UPLC-QTOF–MS analyses” instead of “preliminary and UPLC-QTOF–MS. analysis”

- the Abstract - three sentences start with “Besides”! Please find synonyms

- “UPLC-QTOF–MS” instead of “UPLC-QTOF–M.S.” throughout. See lines 20, 123, 149, 151, 276, 322, 464, 598

- “Table 1 (2, 3, ...)” instead of “table 1 (2, 3, ...). See lines 317, 324, 341, 352, 370, 379, 404, 419, 437, 443, 447

Author Response

Dear Editor,

Thank you very much for your valuable comments to improve the quality of our manuscript. I hope you will consider the corrections regarding our research article entitled “Chemical, pharmacological and in silico studies of stem's extracts of Bauhinia scandens L. provide insights into the management of diarrheal and microbial infections” (Manuscript number: nutrients-1520213) feasibly. We have made corrections following the suggestions of the honorable reviewer. Finally, the corrections were marked with the highlighter. Please see the attachment.

Sincerely yours

Mohammad Aktar Sayeed

Department of Pharmacy

Faculty of Science and Engineering

International Islamic University Chittagong, Chattagram 4318, Bangladesh.

Email: sayeed_ustc@yahoo.com

This manuscript is a resubmission of an earlier submission. The following is a list of the peer review reports and author responses from that submission.